# Epidemiology of Shiga Toxin-Producing *Escherichia coli* Infections in Southern Italy after Implementation of Symptom-Based Surveillance of Bloody Diarrhea in the Pediatric Population

**DOI:** 10.3390/ijerph17145137

**Published:** 2020-07-16

**Authors:** Daniela Loconsole, Mario Giordano, Francesca Centrone, Marisa Accogli, Daniele Casulli, Anna Lisa De Robertis, Anna Morea, Michele Quarto, Antonio Parisi, Gaia Scavia, Maria Chironna

**Affiliations:** 1Department of Biomedical Sciences and Human Oncology-Hygiene Section, University of Bari, Piazza G. Cesare 11, 70124 Bari, Italy; daniela.loconsole@uniba.it (D.L.); francesca.centrone.fc@gmail.com (F.C.); marisa.accogli@uniba.it (M.A.); daniele.casulli@hotmail.com (D.C.); derobertis.annalisa@gmail.com (A.L.D.R.); anna.morea@yahoo.com (A.M.); michele.quarto@uniba.it (M.Q.); 2Pediatric Nephrology and Dialysis Unit, Pediatric Hospital “Giovanni XXIII”, Via Giovanni Amendola, 207, 70126 Bari, Italy; mario.giordano@policlinico.ba.it; 3Istituto Zooprofilattico Sperimentale della Puglia e della Basilicata, Via Manfredonia, 20, 71121 Foggia, Italy; antonio.parisi@izspb.it; 4Food Safety, Nutrition and Veterinary Public Health Department, Istituto Superiore di Sanità, Viale Regina Elena, 299, 00161 Rome, Italy; gaia.scavia@iss.it

**Keywords:** epidemiology, surveillance, Shiga toxin-producing *Escherichia coli*, bloody diarrhea, hemolytic–uremic syndrome, Italy

## Abstract

Shiga toxin-producing *Escherichia coli* (STEC) infections result in a significant public health impact because of the severity of the disease that, in young children especially, can lead to hemolytic–uremic syndrome (HUS). A rise in the number of HUS cases was observed in the Apulia region of Italy from 2013 to 2017, and so, in 2018, a symptom-based surveillance system for children with bloody diarrhea (BD) was initiated in order to detect and manage STEC infections. The objective of the study was to describe the epidemiology of STEC infections in children from June 2018 to August 2019. Children <15 years old with BD were hospitalized and tested for STEC. Real-time PCR for virulence genes (*stx1*, *stx2*, *eae*) and serogroup identification tests were performed on stool samples/rectal swabs of cases. STEC infection was detected in 87 (10.6%) BD cases. The median age of STEC cases was 2.7 years, and 60 (68.9%) were <4. Of these 87 cases, 12 (13.8%) came from households with diarrhea. The reporting rate was 14.2/100,000, with the highest incidence in cases from the province of Bari (24.2/100,000). Serogroups O26 and O111 were both detected in 22/87 (25.3%) cases. Co-infections occurred in 12.6% of cases (11/87). Twenty-nine STEC were positive for *stx1*, *stx2*, and *eae*. Five cases (5.7%) caused by O26 (n = 2), O111 (n = 2), and O45 (n = 1) developed into HUS. A risk-oriented approach based on the testing of children with BD during the summer may represent a potentially beneficial option to improve the sensitivity of STEC surveillance, not only in Italy but also in the context of Europe as a whole.

## 1. Introduction

Shiga toxin-producing *Escherichia coli* (STEC) infections in humans are responsible for causing severe gastrointestinal symptoms, including watery diarrhea and hemorrhagic colitis, and they may also potentially progress to produce severe systemic disorders. STEC infections represent a worldwide public health issue, and the World Health Organization (WHO) estimated that, in 2010, foodborne STEC infections amounted to more than one million cases and 100 deaths globally [1]. 

In 10–15% of patients, STEC infections may result in hemolytic–uremic syndrome (HUS), which is characterized by microangiopathic hemolysis, platelet destruction, and renal failure [2,3]. This condition is mainly triggered by the effects of the Shiga toxin released by STEC in the endothelial cells of target organs, mainly the kidneys. HUS affects children in particular, with a peak of incidence in children <5 years of age, and it shows a case-fatality rate ranging from 3% to 5% [4]. It is the most common cause of acute renal failure in young children and leads to neurological involvement in 25% of cases [3]. Inducing early volume expansion after the occurrence of typical prodromal symptoms of STEC infection represent an opportunity to reduce organ damage in patients at risk of developing HUS, because hemoconcentration and dehydration are considered risk factors for the evolution of HUS [5,6,7]. Transmission pathways of STEC infection include the fecal–oral, foodborne, environmental, and person-to-person routes [8]. The latter, in particular, occurs especially in childcare facilities [9].

Serotype O157:H7 STEC is considered to be most commonly associated with incidents of disease [10,11,12] and to cause severe infections [2]. Nonetheless, as reported for many countries, non-O157 STEC serogroups also have the same pathogenic potential [13]. 

HUS is frequently associated with STEC strains carrying the *stx2* and *eae* genes, and this gene composition is considered high risk for the development of HUS [11,14]. In recent years, outbreaks caused by non-O157 STEC infections with this pattern of virulence genes have been reported in Europe [15,16,17,18]. 

STEC infections are mandatorily notifiable in most EU/EEA (European Economic Area) countries, with the exception of five (France, Luxemburg, Spain, Italy, and the United Kingdom), and surveillance systems have national coverage in all countries except for France, Italy, and Spain [19]. Furthermore, most EU/EAA countries have a passive surveillance system, and only five have laboratory-based reporting of STEC infections. 

In Italy, the STEC surveillance system is primarily based on pediatric HUS cases [19] and, consequently, the occurrence of STEC infection is underestimated [18,20]. During the 1988–2018 surveillance period, HUS cases steadily increased, and non-O157 STEC infections, in particular O26, emerged as the most prevalent serogroups [21]. 

In the Apulia region (Southern Italy), the mean annual reporting rate of HUS in the pediatric population (0.67 HUS cases/100,000 population) was among the highest rates observed in Italy, and the number of HUS cases reported between 2008 and 2017 was 4.6 times higher than in the previous decade (Scavia, personal communication, 2019). This rise was caused by both a severe outbreak of STEC O26 infections in 2013 that caused 20 cases of HUS in children, with many of them developing long-lasting sequelae [18], and higher than expected numbers of sporadic HUS cases occurring between January 2017 and June 2018 (data from Regional Epidemiological Observatory and National Registry of HUS). Three deaths resulting from HUS and its complications caused by STEC infections occurred in the same period. The deaths of these very young children (<2 years) had a dramatic impact on the local media. The recent increase in the number of HUS cases in Apulia has made it necessary to implement strategies to improve the prevention and control of STEC infections and HUS. The implementation of symptom-based surveillance of bloody diarrhea (BD) in the pediatric population was therefore established in 2018 through an operating protocol [22].

The purpose of the surveillance was to both improve the sensitivity of STEC monitoring in the pediatric population, in order to allow the early identification of patients at risk of developing HUS and to detect STEC outbreaks, and to contribute to the characterization of the epidemiology of STEC infections in children. In this study, we report the results of the surveillance of STEC infections causing bloody diarrhea (BD) in children in Apulia from June 2018 to August 2019.

## 2. Materials and Methods

### 2.1. Surveillance Setting

Apulia is a populous region of Southern Italy, with more than four million inhabitants, of whom 527,894 are children <15 years old (Source demographic data: ISTAT, 2019; http://demo.istat.it/pop2018/index.html). In the region, there are 24 hospitals with pediatric units: seven in the province of Bari, three in the province of Barletta-Andria-Trani (BT), two in the province of Brindisi, four in the province of Foggia, five in the province of Lecce, and three in the province of Taranto. 

### 2.2. Epidemiological Surveillance and Operating Protocol for Management of Bloody Diarrhea

A STEC infection symptom-based surveillance system for the pediatric population was established in Apulia in June 2018. The population under surveillance included children <15 years presenting with BD, which was defined as any amount of blood in the loose stool when examined or reported by parents during the previous 2 weeks [22]. According to the protocol, all pediatric units in regional hospitals had to admit all children presenting with BD, conduct a prompt diagnosis of STEC infection, and prevent complications, as suggested in previous studies [5,6,7].

Patients with BD who were identified by primary care pediatricians or by public pediatric outpatient services were immediately hospitalized in the nearest hospital with a pediatric unit. Stool samples/rectal swabs were collected promptly after hospitalization and submitted to the laboratory for STEC testing. In addition, both blood count and renal functions were assessed according to the operating protocol [22]. Personal data and information on the clinical course of the illness and treatment, including antimicrobials administered before hospitalization, were collected for each case. 

Examination of biological specimens for STEC was centralized at the Regional Reference Laboratory of Molecular Epidemiology and Public Health of the Hygiene Unit of the Azienda Ospedaliero-Universitaria Policlinico University Hospital of Bari. Where a STEC infection was identified, the patient’s household contacts were invited, on a voluntary basis, to submit stool samples for STEC testing, even in the absence of symptoms.

### 2.3. Laboratory Diagnosis of STEC Infections

Stool samples and rectal swabs underwent molecular detection of STEC. Total nucleic acid was extracted using the MagnaPure LC automated extraction system (Roche Diagnostics, Milan, Italy). A commercial multiplex real-time PCR kit (FTD, Bacterial gastroenteritis, FastTrack Diagnostics, Luxembourg) was then used for the detection of the *stx1/2* genes. The results of the laboratory test were communicated to the pediatric hospitals within 24 hours of the arrival of fecal samples.

Samples that were positive for the presence of STEC were subsequently tested for the presence of the *stx1*, *stx2*, and *eae* virulence genes by a real-time PCR test (PATHfinder *E. coli* VTEC *stx1-stx2* & *eae*-IAC Duplex Assay, Generon, San Prospero, Modena, Italy). When the molecular test for the virulence genes yielded a negative result, the multiplex real-time PCR for the detection of *stx1/2* genes was repeated, and if the presence of *stx1/2* genes was confirmed (cycle threshold <38), the sample was classified as positive for STEC. Serogroup identification was performed on samples by a commercial real-time PCR test that identifies serogroups O26, O45, O103, O104, O111, O121, O145, and O157 (STEC Identification LyoKit, BIOTECON Diagnostics GmbH, Potsdam, Germany). 

### 2.4. Data Analysis

Data analysis was performed on the demographic and clinical information collected through the surveillance form. Continuous data are reported as medians with the interquartile range (IQR). Analysis of data was performed with STATA 12.0 software (StataCorp LLC, College Station, TX 77845-4512, USA). 

### 2.5. Ethical Statement

The study was not submitted for approval by a research ethics committee because the activities described were conducted as part of the legislated mandate of the Health Promotion and Public Health Department of Apulia, the competent authority for the surveillance of communicable diseases according to Nota Prot. AOO_005_000221 of 21 June 2018, which created the epidemiological surveillance network for BD in Apulia. All activities undertaken formed part of public health surveillance and did not require informed consent. Informed written consent was obtained from all household contacts, or the legal guardians of contacts, of STEC cases who provided samples for STEC detection.

## 3. Results

Between 22nd June 2018 and 21st August 2019, 823 cases of BD in children were identified. Of these, 781 (94.9%) were residents and 42 (5.1%) were non-residents of Apulia. Of the total cases, 55.5% (n = 457) were male and 44.5% (n = 366) female. The median age of resident cases was 3.3 years (IQR: 1.5–7.0). The reporting rate of BD in the resident population was 147.9 per 100,000 children <15 years old. 

A STEC infection was detected in 87 (10.6%) of the BD cases. A clear seasonality in the distribution of STEC infections by month was observed during the surveillance period (Figure 1), with STEC positive cases peaking in the summer. 

The geographical distribution of cases also showed differences, with the greatest number of STEC positive cases occurring in the province of Bari (n = 40; 46% of all STEC positive cases). Altogether, STEC positive cases from all the other Apulia provinces (Lecce, n = 15; Foggia, n = 12; BT, n = 8; Brindisi, n = 7; Taranto, n = 5) accounted for 47/87 STEC cases (54%). Of the STEC cases, 50.6% (44/87) were females, and the median age was 2.7 years (IQR: 1.6–5.9). Sixty cases were in children from 0 to 4 years old (68.9%) and six in infants <1 year old. Fever was reported in 34.5% of cases (n = 30) and vomiting in 25.3% (n = 22). Households with diarrhea were associated with 13.8% of cases (n = 12). Antibiotic use was reported in 10/87 cases (11.5%).

The mean annual reporting rate of cases with STEC infection estimated for the whole surveillance period was 14.2/100,000 in children <15 years old in the Apulia region, with the highest value obtained for the province of Bari (20.6/100,000) (Figure 2). In Apulia, the incidence for children aged 0–1 years was 42.5/100,000, and for those who were 2–4 years old, it was 28.3/100,000 (Figure 3).

Among all serogroups identified (n = 98), comprising co-infections, the most prevalent serogroups were O26 and O111 (both n = 24, 25.5%), followed by O157 (n = 19, 19.4%) and O145 (n = 9, 9.2%) (Figure 4). For 11 children, infection with STEC belonging to two serogroups was identified (O111/O103, O111/O145, O111/O45, O145/O45, O157/O104, O157/O145, O26/O157, and O26/O45 in one case each and O26/O111 in three cases). 

Although positive by the molecular screening test for *stx1/stx2*, 11/87 samples produced a negative result in the molecular test used to discriminate the *stx1* and *stx2* genes. These discrepancies in results could be explained by a different sensitivity in the target detection of the molecular methods used. In Table 1, the distribution of serogroup and virulence gene composition for samples from 76 cases of STEC infection is reported. The presence of *stx1* + *stx2* was found in samples from 31/76 cases (40.8%). All three main virulence genes (*stx1*, *stx2*, *eae*) were detected in 29 cases (38.2%), mainly in samples that tested positive for serogroups O111 and O157. 

In the whole period of surveillance, five children (5.7%) with STEC infections developed HUS. In addition, six cases not detected by surveillance occurred (one of these cases was resident outside of the region). The mean annual reporting rate of HUS cases (n = 11) in the period considered was 1.8/100,000. In 2018, two STEC-associated HUS cases occurred in female twins of 18 months of age, and these were caused by STEC O26; one case in a 7-year-old female was caused by STEC O45; and one case in a 9-year-old female was caused by STEC O111. The only case that occurred in 2019 was in a 2-year-old female and was caused by STEC O111. All STEC-associated HUS cases were positive for the virulence genes *stx1*, *stx2*, and *eae.*

In 18/87 (20.7%) cases of STEC infection, samples from household contacts were collected. Investigation of STEC cases’ family contacts allowed the identification of ten further STEC infections (one with diarrhea and nine asymptomatic), linked to seven STEC-positive children with BD. All ten positive household contacts were older than the cases. Five contacts were adults and five were siblings. Samples from all but one STEC-positive household showed the same serogroup and virulence genes as the cases.

Starting from the activation of the surveillance, it seems that no outbreaks of HUS occurred. 

## 4. Discussion

This study describes the epidemiology of STEC infections in children with BD in Southern Italy after the implementation of a symptom-based surveillance program. During the study, the reporting rates for STEC infections and HUS cases in the pediatric population were 14.2 and 1.8 cases/100,000, respectively, amounting to a ratio of almost eight children with BD and STEC infection for every single case of pediatric HUS. 

Compared with the mean notification rate of STEC infections in the EU/EEA in 2018 of 2.3 cases/100,000 population [23], the reporting rate of STEC infection estimated in our study is more than six times higher. It is only exceeded by Ireland (20.0 cases/100,000), which is the EU country with the highest notification rate over the last 5 years. Besides true epidemiological differences, the higher reporting rate estimated in Apulia is, we suggest, attributable to the active testing of symptomatic cases presenting with BD, although other reasons may also contribute to this finding. Our study only focused on pediatric cases (<15 years old), which is the population at the highest risk for STEC infection. In addition, the period of surveillance in Apulia mainly included the warmer seasons, when STEC infections are at their highest. While these factors may introduce a bias in the comparability of our findings with the estimates at the EU/EEA level, which are for the whole population and are not seasonally biased, the results of this study can be interpreted as reflecting results obtained by improving the sensitivity of a surveillance program. Our results suggest that a risk-oriented approach, based on the active testing of children with BD, especially during the risk season, may be a potentially beneficial option for improving the sensitivity of STEC surveillance.

During the surveillance period, STEC infection was identified in approximately 11% of BD cases. This finding is consistent with 9% being reported in a one-year cohort study conducted recently in a single hospital, where 108 cases of BD were studied [24]. Most cases of STEC infection, irrespective of serogroup, occurred in the warmer months, thus agreeing with other studies that also found a seasonal trend for STEC infections [19,25].

The province of Bari showed the highest reporting rate of STEC infections during the surveillance period. This finding could reflect the real incidence of STEC infections in this province or an under-notification of BD cases by hospitals in other provinces in the first few months of the surveillance, as well as the failure to send fecal samples for testing because of the distance from the Regional Reference Laboratory. Notably, HUS cases detected in recent years in Apulia were mainly reported in the province of Bari. Furthermore, the majority of HUS cases occurred during a 2013 outbreak that occurred in the same province [18]. It could be hypothesized that the population of this province is more frequently exposed to risk factors for STEC infections, such as a wider environmental circulation of the pathogen or a higher frequency of consumption of contaminated food. 

Data produced in the present study show that 70% of STEC positive cases in the pediatric population occurred in the 0–4 years age group. In the EU, almost one-third of all confirmed STEC cases have been reported in children aged <4 years, which is the age group with the highest reporting rate (8.9 cases/100,000 population), confirming that very young children are at a higher risk of infection than people of other ages [19,26,27]. 

In the EU, O157 is the most frequently detected serogroup, but in recent years, the emergence of non-O157 serogroups has been reported [19]. This is possibly an effect of the increasing awareness of non-O157 STEC, as well as the fact that more laboratories commenced testing for serogroups other than O157 [28]. Overall, non-O157 STEC infections accounted for more than 80% of the total, and serogroups O26 and O111 were the most frequently occurring serogroups in Apulia. In Italy, the most common non-O157 serogroups that were reported were also O26 and O111 (National Registry of HUS). In addition, these findings agree with the estimated proportion of non-O157 STEC infections in Germany, which accounted for approximately 80% of all gastroenteritis cases caused by STEC and half of all BD caused by STEC [29]. Even in countries outside Europe, non-O157 serogroups are becoming more commonly detected than serogroup O157. For example, in Canada, the estimated proportion of non-O157 serogroups causing gastroenteritis was 62% [30], while in the United States, it was 64% [31]. 

Shiga toxins produced by the *stx1* and *stx2* genes are the main virulence factors of STEC [32], whereas the *eae* gene, encoding for intimin protein, mediates the attachment of STEC to the enterocytes [33]. It is known that *stx2* and *eae* genes, along with young age, play the roles of risk factors for the development of HUS [11,14,27]. In our study, samples from 32.9% (25/76) of STEC cases harbored the *stx2* gene (single and co-infection). It has been suggested that infections caused by *E.coli* strains carrying the *stx2* gene are more likely to result in HUS than those caused by STEC carrying both *stx1* and *stx2*, thus indicating that *stx1* could mitigate, to some extent, the effect of the *stx2* gene [14]. However, the five HUS cases observed in Apulia during the surveillance period, who were previously diagnosed with a STEC infection, as well as 3/6 further HUS cases registered in the region, were all caused by STEC that were positive for the *stx1*, *stx2* and *eae* genes. This difference could be related to the fact that, in the previous study, approximately 50% of STEC infections were caused by serogroup O157 and were found in all age groups, whereas our surveillance targeted only the pediatric population, and non-O157 STEC were the prevalent serogroups involved with infections. 

During the surveillance period, <6% of cases detected and diagnosed as STEC-positive developed HUS, which is lower than the 15% reported by Tarr et al. and 20% by Adams N et al., although the latter data relate to STEC O157 infections only [2,27]. Not all the patients with STEC infection develop BD, and this finding may partly explain this difference. Moreover, very few data are available regarding the development into HUS of non-O157 STEC infections or, in particular, regarding the emerging O26 and O111 STEC serogroups, which represented more than 50% of the STEC infections in the present study. In the 2011 community-wide outbreak caused by STEC O104 in Germany, HUS occurred in 22% of cases [16], thus indicating that non-O157 STEC infections could more frequently lead to complications. However, the outbreak strain was considered to be a hybrid possessing an unusual combination of pathogenic features typical of enteroaggregative *E. coli*, together with the capacity to produce the Shiga toxin. 

The risk of developing HUS could be associated with genetic factors [34]. This aspect could be hypothesized in the two female twins with STEC O26 infection but, unfortunately, this issue was not investigated.

STEC are zoonotic agents, and human infections are mostly related to foodborne exposure, but person-to-person spread is also common due to the possibility for infections to occur even at low doses [35,36]. In our study, ten STEC infections were identified in the household contacts of STEC cases, probably indicating an exposure to the same source of infection or person-to-person transmission. The latter transmission route is common in childcare settings [13,37]. Further investigations into the carriage by, or infection of, household contacts are needed in order to allow a better evaluation of the role of person-to-person transmission within family settings and to elucidate further the sources of infection in young children and infants.

This study contributes to an improved understanding of the epidemiological framework of STEC infections in Southern Italy, but it has some limitations. Firstly, the implementation of the surveillance program during the startup phase in the summer of 2018 could have been affected by a lack of dissemination of the protocol among pediatricians. Secondly, pediatric units of more distant provinces from the Regional Reference Center for STEC diagnosis and surveillance may not have sent fecal samples from all hospitalized children with BD, and this could explain differences in the reporting rates in some provinces of the region. Thirdly, no statistical analysis was performed in the study to compare data or to assess local risk factors that can influence the probability of STEC infection among BD patients. Taken together, these factors may have affected the efficiency of STEC infection detection. Nonetheless, the surveillance established in Apulia can be considered as a pilot project and provides preliminary experience in exploring ways to obtain robust data on this emerging public health problem. 

The increasing number of HUS cases in Apulia over recent years presents a concern about the negative impact on regional productivity, as Apulia is a leading region for dairy products, a food commodity thought to be involved in the 2013 outbreak [18]. STEC infections are often travel-related, with many children developing HUS after returning from typical tourist destinations. Therefore, this experience is also relevant in view of the importance of the region as a tourist destination. Our symptom-based surveillance of BD in children could represent a valid model for enhancing STEC surveillance at a national level, even for limited time periods (e.g., peak STEC season). The implementation of a nationwide surveillance program and the application of strict preventive measures could help to reduce the impact of STEC infections. 

## 5. Conclusions

Continuation of the current surveillance of BD in children and the application of the operating protocol established in Apulia will help to clarify many cryptic aspects of STEC infection, such as the burden of the disease. Additionally, the molecular characterization of STEC strains could provide valuable further information, particularly in outbreak investigations. Finally, more effective preventive measures should be enforced in order to control STEC carriage in animals and its spread to the environment [31], since a one-health approach is needed in order to reduce the occurrence of outbreaks.

## Figures and Tables

**Figure 1 ijerph-17-05137-f001:**
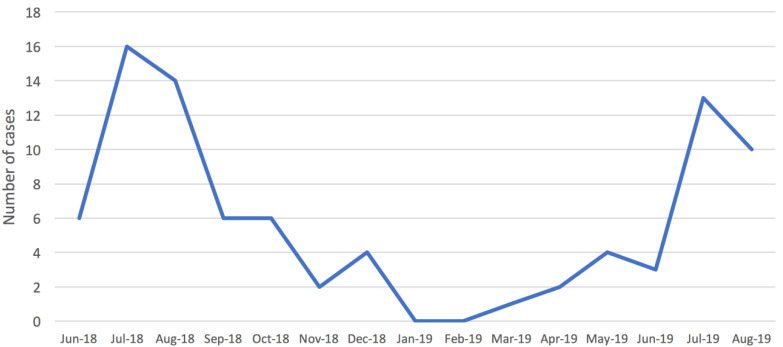
Distribution of confirmed cases of Shiga toxin-producing *Escherichia coli* (STEC) infection in children (<15 years old) by month, Apulia region, June 2018–August 2019.

**Figure 2 ijerph-17-05137-f002:**
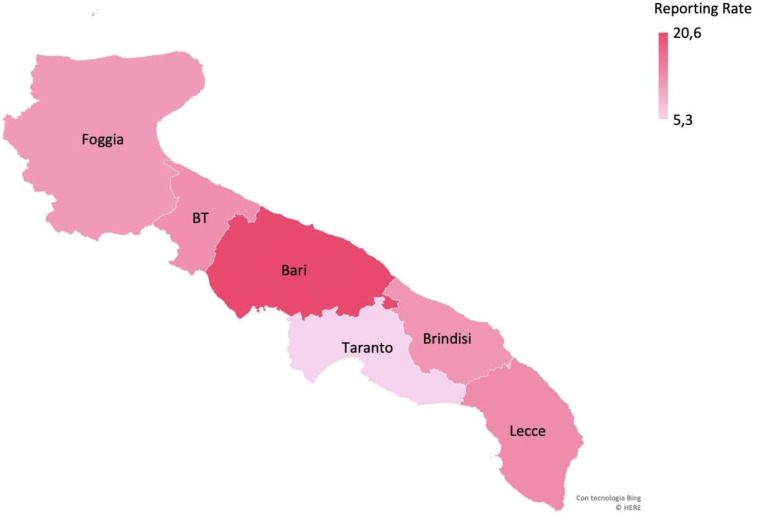
Mean annual reporting rate (per 100,000 children <15 years old) of cases of STEC infection by province, Apulia region.

**Figure 3 ijerph-17-05137-f003:**
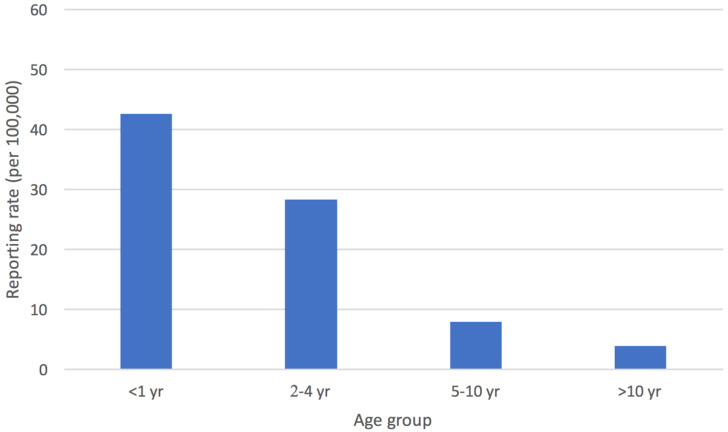
Mean annual reporting rate (per 100,000 children <15 years old) of cases of STEC infection by age group, Apulia region.

**Figure 4 ijerph-17-05137-f004:**
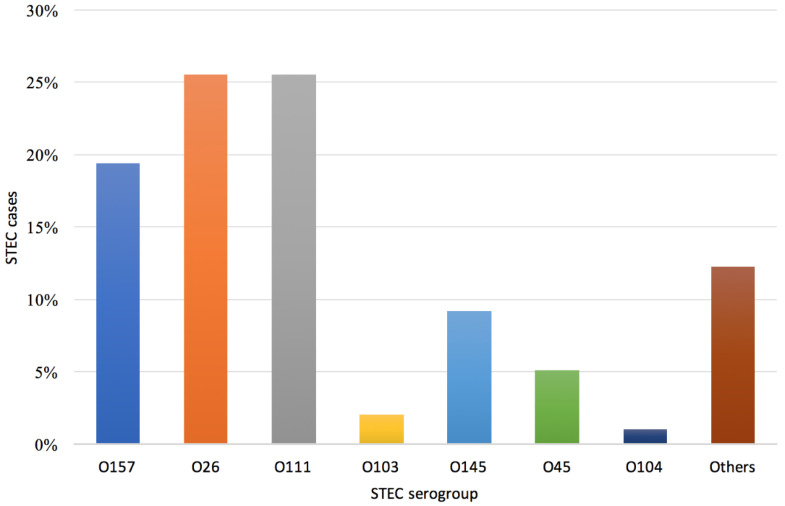
Distribution of STEC serogroups (n = 98) detected in children with bloody diarrhea, Apulia region, June 2018–August 2019.

**Table 1 ijerph-17-05137-t001:** Distribution of cases of STEC infection with identified virulence pattern (n = 76) by STEC serogroup and virulence genes, Apulia region, July 2018–August 2019.

		Virulence Genes			
No. of Cases	Serogroup	*stx1*	*stx2*	*stx1 + stx2*	*eae*
**18**	**O26**				
	n = 1	-	-	+	+
	n = 14	+	-	-	+
	n = 2	-	+	-	+
	n = 1	-	+	-	-
**17**	**O111**				
	n = 14	-	-	+	+
	n = 2	-	+	-	+
	n = 1	-	+	-	-
**15**	**O157**				
	n = 11	-	-	+	+
	n = 4	-	+	-	+
**6**	**O145**				
	n = 2	-	-	+	+
	n = 2	+	-	-	+
	n = 2	-	+	-	+
**1**	**O103**				
	n = 1	+	-	-	+
**9**	***Co-Infections***				
	**O111/O26** n = 2	+	-	-	+
	**O111/O45** n = 1	-	-	+	+
	**O111/O103** n = 1	-	+	-	+
	**O145/O45** n = 1	-	+	-	+
	**O157/O104** n = 1	-	+	-	+
	**O157/O145** n = 1	-	+	-	+
	**O26/O157** n = 1	-	+	-	+
	**O111/O145** n = 1	-	-	+	-
**10**	***Other Serogroups***				
	n = 1	+	-	-	+
	n = 5	-	+	-	+
	n = 1	-	-	+	-
	n = 3	-	+	-	-

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
