# Peer review of "Epidemiology of Shiga Toxin-Producing Escherichia coli Infections in Southern Italy after Implementation of Symptom-Based Surveillance of Bloody Diarrhea in the Pediatric Population"

_ijerph, 2020, doi:10.3390/ijerph17145137_

Round 1

Reviewer 1 Report

I may say that the present study is quite interesting to read. I have some questions at the beginning, but they were resolved mainly while reading the Discussion. I liked the story of O157 and non-O157 STEC infections and attempt of the authors to highlight the importance of detections of non-O157. I have only few remarks below and recommend the paper for acceptance with minor revision (I wish to see the response on how false negatives/positives were handled by the authors).   L190: if the HUS is quite rare and two cases in 2018 occurred in two female twins, was it some explanation for this? For example, was any possible genetic factor involved? L190: I also wonder, is there any age-correlation in HUS? For example, only very young or older children are likely to acquire HUS? General: Is it possible to have false positive/negatives in the tests? How is this issue usually addressed? 

Author Response

Reviewer #1

l may say that the present study is quite interesting to read. l have some questions at the beginning, but they were resolved mainly while reading the Discussion. I liked the story of 0157 and non-0157 STEC infections and attempt of the authors to highlight the importance of detections of non-0157. I have only few remarks below and recommend the paper for acceptance with minor revision (l wish to see the response on how false negatives/positives were handled by the authors).

Response: We thank the Reviewer for the very encouraging comments.

L 190: if the HUS is quite rare and two cases in 2018 occurred in two female twins, was it some explanation for this? For example, was any possible genetic factor involved?

Response: We thank the Reviewer for the helpful comment. The risk of developing HUS could be associated with genetic factors but, unfortunately, this issue was not investigated for the two twins developing HUS. We have added a sentence (page 9, lines 291-293) and a reference to make explicit this aspect (Frémeaux-Bacchi V, et al. Clin J Am Soc Nephrol. 2019, reference 34).

L 190: l also wonder, is there any age-correlation in HUS? For example, only very young or older children are likely to acquire HUS? General: ls it possible to have false positive/negatives in the tests? How is this issue usually addressed?

Response: We thank the Reviewer for the comment. HUS is more frequent in children, with a peak of incidence in children <5 years of age. We have added a sentence in the “Introduction” section (page 2, lines 55-56) and a reference (Fitzpatrick M. Haemolytic uraemic syndrome and E. coli O157. BMJ. 1999, reference 4).

False or negative results could occur. However, false-positive results in symptomatic patients are very rare. Nevertheless, to avoid a false-positive result, all samples positive at the multiplex real-time PCR for the detection of stx1/2 genes but negative at the molecular test used to discriminate the stx1 and stx2 genes were repeated. If the presence of stx1/2 genes was confirmed (cycle threshold <38), the sample was classified as positive for STEC. We have now added a sentence to clarify this aspect in the “Materials and methods” section (page 3, lines 131-134). A difference in the sensitivity of the molecular methods used could explain the different results. We have made explicit this aspect in the “Results” section (page 6, lines 191-193). False-negative results are improbable since “internal controls” were used to verify the adequacy of the sample, the effectiveness of the nucleic acid extraction, and the amplification process.

Reviewer 2 Report

Manuscript Title: Epidemiology of Shiga toxin-producing Escherichia  coli infections in Southern Italy after implementation  of symptom-based surveillance of bloody diarrhea in the pediatric population

I commend the authors for their study on the specified Escherichia coli infections in Southern Italy. However, the manuscript is inadequate on multiple levels.

  1. The background needs to be expanded to provide historical trends, social, epidemiological and environmental trends in the prevention and control of the specified Escherichia coli infections, including how widely the current protocol is used, if any, in the wider region, and the experience of other provinces in Italy or other countries in Europe in dealing with this particular public health problem.
  2. The sample consists of symptomatic cases that made pediatric visits to health care facilities. Do the authors think that the sample is representative of the source population? If cases are not representative, then the validity of the study is uncertain.
  3. Does the study accomplish its purpose?

Page 2: lines 83-86.“The purpose of the surveillance was to improve both the sensitivity of STEC monitoring in the pediatric population to allow the early identification of patients at risk of developing HUS and to detect STEC outbreaks, as well as to contribute to the characterization of the epidemiology of STEC  infections in children.”  If this is the purpose of the study, was it accomplished by the study? How much was the improvement in sensitivity compared to prior protocols? Was there any improvement in the early detection of outbreaks? Was there any improvement in identifying most at-risk patients?

  1. What parameters qualify this report as a well-designed study?
  2. What is/are the research question(s)?
  3. What is presented as the statistical analysis does not uphold acceptable standards. PagePage 5: lines 170-174. “Among all serogroups identified (n=98), comprising co-infections, the most prevalent  serogroups were O26 and O111 (both n=24, 25.5%), followed by O157 (n=19, 19.4%) and O145 (n=9,  9.2%) (Figure 4). For 11 children, infection with STEC belonging to two serogroups was identified  (O111/O103, O111/O145, O111/O45, O145/O45, O157/O104, O157/O145, O26/O157, and O26/O45 in one case each, and O26/O111 in three cases).,

The authors are comparing percentages from their descriptive statistics. In statistics and epidemiology,  it is not possible to definitively state 25.5% is greater than 19.4% without doing the one-sample t-test or the two-sample t-test, etc. at your chosen confidence level and p-value.

  1. The study looks like a routine public health surveillance report for local or national consumption in Italy rather than well-designed research. The results need to quantify the improvements the current approach used in the investigation of the epidemic compared to previous methods
  2. The authors need to provide coherent evidence to convince the readers that their study expands our understanding, bridges a definite knowledge gap, or provided new knowledge that influences policy and public health program interventions in Italy and beyond.

Author Response

I commend the authors for their study on the specified Escherichia coli infections in Southern

ltaly. However, the manuscript is inadequate on multiple levels.

1.The background needs to be expanded to provide historical trends, social, epidemiological and environmental trends in the prevention and control of the specified Escherichia coli infections, including how widely the current protocol is used, if any, in the wider region, and the experience of other provinces in ltaly or other countries in Europe in dealing with this particular public health problem.

Response: We thank the Reviewer for the comment. Hemolytic-Uremic Syndrome (HUS) is a complication of Shiga toxin-producing E. coli infections and represents a public health problem in European countries, comprising Italy. Also, the prevention and control of STEC infections is a challenge. In 2018, the Apulia region has set up symptom-based surveillance of STEC infections in the pediatric population to improve the prevention and control of STEC infections and HUS (“Introduction” section, page 2, lines 87-90). To the best of our knowledge, there are no other experiences based on the activation of a specific operating protocol to manage STEC infections. In Italy, the STEC surveillance system covers only pediatric HUS cases. In EU/EAA, STEC infections are not uniformly reported and most countries have passive surveillance systems. We have now expanded the background as suggested (page 2, lines 70-77).

2. The sample consists of symptomatic cases that made pediatric visits to health care facilities.

Do the authors think that the sample is representative of the source population? lf cases are not representative, then the validity of the study is uncertain. Does the study accomplish its purpose?

Page 2: lines 83-86."The purpose of the surveillance was to improve both the sensitivity of STEC monitoring in the pediatric population to allow the early identification of patients at risk of developing HUS and to detect STEC outbreaks, as well as to contribute to the characterization of the epidemiology of STEC infections in children." lf this is the purpose of the study, was it accomplished by the study? How much was the improvement in sensitivity compared to prior protocols? Was there any improvement in the early detection of outbreaks? Was there any improvement in identifying most at-risk patients?

Response: Thank you. Symptom-based surveillance is mandatory in the Apulia region. Therefore, all pediatric cases of bloody diarrhea should be hospitalized according to the operating protocol. The surveillance covers the entire region and, therefore, the sample is representative of the source population. The purpose was accomplished by the study since there was no systematic data on STEC infections in the pediatric population before the activation of this surveillance. Moreover, the early detection of STEC infections allowed the identification of children at risk of developing HUS and the application of early volume expansion to prevent HUS and/or long-lasting sequelae. Starting from the activation of the surveillance, it seems that no outbreaks occurred. A sentence has been added in the “Results” section (page 7, line 214).

4. What parameters qualify this report as a well-designed study?

5. What is/are the research question(s)?

Response: We thank the Reviewer for the comments. The data reported in the manuscript are the results of a symptom-based surveillance system. This surveillance gave us the opportunity to identify children with bloody diarrhea caused by STEC infections. These data were not available before.

6. What is presented as the statistical analysis does not uphold acceptable standards.

Page 5: lines 170-174. "Among all serogroups identified (n=98), comprising coinfections, the most prevalent serogroups were 026 and 0111 (both n=24, 25.5%), followed by 0157 (n=19, 19.4%) and 0145 (n=9, 9.2%) (Figure 4). For 11 children, infection with STEC belonging to two serogroups was identified (0111 /0103, 0111/0145, 0111 /045, 0145/045, 0 157/0104, 0157/0145, 026/0157, and 026/045 in one case each, and 026/0111 in three cases). The authors are comparing percentages from their descriptive statistics. In statistics and epidemiology, it is not possible to definitively state 25.5% is greater than 19.4% without doing the one-sample t-test or the two-sample t-test, etc. at your chosen confidence level and p-value.

Response: We thank the Reviewer for the comment. We are aware that the comparison among percentages cannot be considered “significant”. In “Data analysis” paragraph we never stated that we performed “statistical analysis”. STATA 12.0 was only used for the descriptive analysis. Nevertheless, we reported the lack of statistical analysis as a limit of the study in the Discussion section (page 9, lines 308-310).

8.The study looks like a routine public health surveillance report for local or national consumption in ltaly rather than well-designed research. The results need to quantify the improvements the current approach used in the investigation of the epidemic compared to previous methods

Response: Thank you. Yes, the study reports the findings of public health surveillance. The aim of the study was to gain knowledge on the epidemiology of a disease of which little is known. We think the goal was achieved. It should be noted that no systematic data on STEC epidemiology in Italy are available since STEC infection is not routinely notified.

Only continuing the surveillance and appropriate management of STEC infection will allow evaluating if this enhanced symptom-based surveillance will contribute to reducing the incidence of HUS in our region and if it could represent a possible model for this public health issue.

9. The authors need to provide coherent evidence to convince the readers that their study expands our understanding, bridges a definite knowledge gap, or provided new knowledge that influences policy and public health program interventions in ltaly and beyond.

Response: We thank the Reviewer for the comment. The novelty of the study consists of the description of the epidemiology of STEC infections among children with bloody diarrhea. Before the starting of the symptom-based surveillance in the Apulia region, no systematic data were available on the circulation of STEC in children. Also in Italy, there are no data on STEC infections. Only data on HUS are available from the Italian HUS Registry. The study expands our understanding of many cryptic aspects of STEC infections, such as the burden of the disease.

Reviewer 3 Report

Even though in general terms manuscript present a high quality information, novelty and important insights about the behavior/epidemiology of STEC and the generation of HUS in high risk group, limited statistical procedures and low quality figures affects the correct use of the information described.

It is important that authors correct the quality of figures and correctly declare what data analysis mean, no analytical tools were declared in the manuscript further than descriptive statistics. Deeper analysis could be done with data, establishing local risk factors that determine the probability of STEC positivity among BD patients.

Comments are included in .pdf file.

Author Response

Even though in general terms manuscript present high quality information, novelty and important insights about the behavior/epidemiology of STEC and the generation of HUS in high risk group, limited statistical procedures and low quality figures affects the correct use of the information described.

lt is important that authors correct the quality of figures and correctly declare what data analysis

mean, no analytical tools were declared in the manuscript further than descriptive statistics. Deeper analysis could be done with data, establishing local risk factors that determine the probability of STEC positivity among BD patients.

Response: We thank the Reviewer for the helpful comments. Figures with high resolution have been provided. No statistical analysis was performed in the study. STATA 12.0 was only used for the descriptive analysis. We have now clearly indicated this aspect as a limit of the study (page 9, lines 308-310). A deeper analysis will be done in the future to establish local risk factors that determine the probability of STEC infection among BD patients.

Comments are included in .pdf file

Response: All changes have been done

Round 2

Reviewer 2 Report

The authors have addressed the questions and comments by this reviewer. Despite the limitations and in view of the fact that public health surveillance provides new information to facilitate the prevention and control of the disease under consideration, the results of this surveillance as presented in this study could provide useful information for ongoing local public health interventions.